# The Importance of Being Bayesian in Online Conformal Prediction

## Abstract

Based on the framework of *Conformal Prediction* (CP), we study the online construction of valid confidence sets given a black-box machine learning model. Converting the targeted confidence levels to quantile levels, the problem can be reduced to predicting the quantiles (in hindsight) of a sequentially revealed data sequence, where existing results can be divided into two types.

- Assuming the data sequence is iid, one could maintain the empirical distribution of the observed data as an algorithmic belief, and directly predict its quantiles.
- As the iid assumption is often violated in practice, a recent trend is to apply first-order online optimization on moving quantile losses [GC21]. This indirect approach requires knowing the targeted quantile level beforehand, and suffers from certain validity issues on the obtained confidence sets, due to the associated loss linearization.

This paper presents a Bayesian approach that combines their strengths. Without any statistical assumption, it is able to both

- answer multiple arbitrary confidence level queries online, with provably low regret; and
- overcome the validity issues suffered by first-order optimization baselines, due to being "data-centric" rather than "iterate-centric".

From a technical perspective, our key idea is to take the above iid-based procedure and regularize its algorithmic belief by a Bayesian prior, which "robustifies" it by simulating a non-linearized *Follow the Regularized Leader* (FTRL) algorithm on the output. For statisticians, this can be regarded as an online adversarial view of Bayesian nonparametric distribution estimation. Importantly, the proposed belief update backbone is shared by "prediction heads" targeting different confidence levels, bringing practical benefits similar to U-calibration [KLST23].

## 1 Introduction

Modern machine learning (ML) models are better at point prediction compared to probabilistic prediction. For example, when given an image classification task, they are better at responding "*this image is most likely a white cat*", rather than "*I'm 90% sure this image is an animal, 60% sure it's a cat, and 30% sure it's a white cat*". For downstream users, the more nuanced probabilistic predictions are often important for risk assessment. The challenge, however, lies in aligning the model's own uncertainty evaluation with its actual performance in the real world.

*Conformal Prediction* (CP) [VGS05] has recently emerged as a premier framework to address this challenge, as it blends the empirical strength of modern ML with the theoretical soundness of

Submitted to Workshop on Bayesian Decision-making and Uncertainty, 38th Conference on Neural Information Processing Systems (BDU at NeurIPS 2024). Do not distribute.

traditional statistical methods. In a nutshell, CP algorithms make *confidence set predictions* (rather than point predictions) on the label space, by sequentially interacting with three other parties: the *nature* (i.e., the data stream), a *black-box ML model*, and *downstream users*. Writing the covariate-label space as $\mathcal{X} \times \mathcal{Y}$ and the time horizon as $T$, we consider the following sequential interaction protocol. In each (the $t$-th) round,

1. We, as the CP algorithm, observe a *target covariate* $x_t \in \mathcal{X}$ from the nature, and a *score function* $s_t : \mathcal{X} \times \mathcal{Y} \to [0, R]$ generated by a black-box ML model BASE.[1]

2. The downstream users select a finite set of *confidence level queries*, $A_t \subset [0, 1]$. Given each $\alpha \in A_t$, we predict a *score threshold* $r_t(\alpha)$,[2] which leads to a *confidence set*

$$\mathcal{C}_t(x_t, \alpha) = \{y \in \mathcal{Y} : s_t(x_t, y) \geq r_t(\alpha)\}. \tag{1}$$

3. We observe the *ground truth label* $y_t \in \mathcal{Y}$ from the nature, and send the $(x_t, y_t)$ pair to BASE (which it optionally uses to update the score function $s_{t+1}$). Define the *true score* $r_t^* := s_t(x_t, y_t)$.

**Limitation of prior work**  The essential objective of CP is to have the prediction $r_t(\alpha)$ close to the $(1 - \alpha)$-*quantile* of the true score sequence $r_{1:T}^*$, while only knowing $r_{1:t-1}^*$ [Rot22, Tib23]. For the readers' reference, the $(1 - \alpha)$-quantile of a real random variable $X$ is defined as $q_{1-\alpha}(X) := \min\{x; \mathbb{P}(X \leq x) \geq 1 - \alpha\}$. Guided by this general principle, the community has focused on two very different approaches under distinct assumptions.

- Assuming the sequence $r_{1:T}^*$ is iid, it suffices to maintain the empirical distribution of $r_{1:t-1}^*$, denoted as $P_t = \bar{P}(r_{1:t-1}^*)$, as an *algorithmic belief*. Then, when queried with the confidence level $\alpha$, the CP algorithm directly "post-processes" the belief by setting $r_t(\alpha) = q_{1-\alpha}(P_t)$, or in situations with only *exchangeability*, $q_{1-\alpha-o(1)}(P_t)$ [Tib23]. This is essentially *Empirical Risk Minimization* (ERM) with the quantile loss $l_\alpha(r, r^*) := (\alpha - \mathbf{1}[r < r^*])(r - r^*)$, i.e.,

$$r_t(\alpha) = q_{1-\alpha}(P_t) \in \arg\min_{r \in [0,R]} \sum_{i=1}^{t-1} l_\alpha(r, r_i^*). \tag{2}$$

- Since the iid assumption is often violated in practice, a recent trend [GC21] is to indirectly view CP as an instance of *adversarial online learning* [Haz23, Ora23], and apply first-order optimization algorithms from there. Taking gradient descent for example, such an approach amounts to picking $r_1(\alpha) \in [0, R]$ and following with the projected incremental update

$$r_{t+1}(\alpha) = \Pi_{[0,R]} \left[ r_t(\alpha) - \eta_t \partial l_\alpha(r_t(\alpha), r_t^*) \right],$$

where $\eta_t > 0$ is the *learning rate*, and $\partial l_\alpha(r, r^*)$ can be any subgradient of the quantile loss $l_\alpha$ with respect to the first argument.

Strictly speaking the two approaches are incomparable due to targeting different performance metrics, but nonetheless, let us compare the *algorithms* side by side. Although first-order optimization seems more robust due to the nonnecessity of statistical assumptions, it requires being "iterate-centric" rather than "data-centric": one needs to fix a single confidence level $\alpha$ beforehand, and the prediction $r_t(\alpha)$ depends on how previous predictions $r_{1:t-1}(\alpha)$ compare to the "data" $r_{1:t-1}^*$, rather than just the "data" itself. This leads to some paradoxical observations regarding the obtained confidence sets. For example,

- The confidence set $\mathcal{C}_t$ is not invariant to permutations of $r_{1:t-1}^*$.

- Suppose one runs two first-order optimization algorithms targeting different $\alpha$ (say, $\alpha_1 < \alpha_2$), then even if the initialization $r_1(\alpha_1) = r_1(\alpha_2)$, it is still possible that $\mathcal{C}_t(x_t, \alpha_1)$ is strictly contained in $\mathcal{C}_t(x_t, \alpha_2)$. That is, the confidence sets violate the monotonicity of probability measures.

In contrast, the ERM approach does not suffer from such issues, therefore is more "valid / plausible" in some sense. The problem is that ERM, also known as *Follow the Leader* (FTL) in online learning, is not robust to adversarial environments with quantile losses. Can we enjoy the best of both worlds?

---

[1]An example is classification, where the score function is usually the softmax score of each candidate label ($R = 1$). It is *positively oriented*: larger means the model is more certain that the candidate label is the true one. For regression, it is more common to use *negatively oriented* score functions, which means the inequality in Eq.(1) is reversed.

[2]This extended abstract focuses on *marginal* CP. More generally, the CP algorithm can predict $r_t(x_t, \alpha)$.

76 **Contribution** This paper presents a Bayesian approach to CP, which ($i$) does not require any
77 statistical assumption; ($ii$) does not suffer from the aforementioned validity issues; and ($iii$) efficiently
78 handles multiple, arbitrary confidence levels revealed online, with provably low regret. Our main
79 workhorse, in short, is an online adversarial view of Bayesian nonparametric estimation.

## 2 Main result

81 **Overview** Our proposed algorithm (Algorithm 1) is perhaps the simplest one could think of.
82 Defining the *Bayesian prior* as an arbitrary distribution $P_0$ on the domain $[0, R]$ (with strictly positive
83 density $p_0 : [0, R] \to \mathbb{R}_{>0}$), we update the algorithmic belief $P_t$ by mixing $P_0$ with the empirical
84 distribution of the previous true scores, $\bar{P}(r^*_{1:t-1})$. This can be seen as regularizing the frequentist
85 belief update $P_t = \bar{P}(r^*_{1:t-1})$. Then, given each queried confidence level $\alpha$, our algorithm picks
86 $r_t(\alpha) = q_{1-\alpha}(P_t)$ just like the iid-based approach. It is clear that $r_t(\alpha)$ is invariant to permutations
87 of $r^*_{1:t-1}$, and for any $\alpha_1 < \alpha_2$ we always have $r_t(\alpha_1) \le r_t(\alpha_2)$.

---

**Algorithm 1** Online conformal prediction with regularized belief.

---

**Require:** Step sizes $\{\lambda_t\}_{t \in \mathbb{N}_+}$, where each $\lambda_t \in [0, 1]$ and $\lambda_1 = 1$. Bayesian prior $P_0$.
 1: **for** $t = 1, 2, \ldots$ **do**
 2:    Compute the empirical distribution $\bar{P}(r^*_{1:t-1})$, and set the algorithmic belief $P_t$ to

$$P_t = \lambda_t P_0 + (1 - \lambda_t) \cdot \bar{P}(r^*_{1:t-1}). \tag{3}$$

 3:    **for** $\alpha \in A_t$ **do**
 4:       Output the score threshold $r_t(\alpha) = q_{1-\alpha}(P_t)$.
 5:    **end for**
 6:    Observe the true score $r^*_t$.
 7: **end for**

---

88 Our central observation, however, is quite profound in our opinion:

89          The Bayesian regularization on the algorithmic belief $P_t$ induces *downstream*
90          *regularizations* on the predicted threshold $r_t(\alpha)$.

91 In particular, Theorem 1 shows that despite not knowing $\alpha$ beforehand, Algorithm 1 generates the
92 same output $r_t(\alpha)$ as a non-linearized *Follow the Regularized Leader* (FTRL) algorithm with the
93 quantile loss $l_\alpha$. To provide more context, FTRL is a standard improvement of ERM / FTL with
94 better stability in adversarial environments, and our framework involves its non-linearized version
95 which retains the full structure of quantile losses. It is also important to note that the *downstream*
96 *simulation* of FTRL deviates from the common scope of online learning (which requires specifying a
97 single loss function in each round [Haz23, Ora23]), and instead has a similar flavor as the recently
98 proposed *U-calibration* [KLST23, LSS24]: forecasting for an *unknown* downstream agent.

99 From a more technical perspective: prior works on U-calibration considered the setting of "finite-class
100 distributional prediction" with generic *proper* losses [KLST23, LSS24], while our paper focuses on
101 the continuous domain $[0, R]$ (i.e., "infinitely many classes") with the more specific quantile losses.
102 The extra problem structure allows our algorithm to be deterministic (rather than *Follow the Perturbed*
103 *Leader*; FTPL), thus establishing a closer connection to deterministic *online convex optimization*.

104 Appendix A further discusses the interpretation of the belief update Eq.(3) as *Bayesian nonpara-*
105 *metric distribution estimation*. The nontrivial insight here is that this statistical procedure induces
106 downstream adversarial regret bounds, without statistical assumptions at all.

107 **Analysis** Formally, we first present the FTRL-equivalence of Algorithm 1, which can be compared
108 to the FTL-equivalence of the iid-based approach, i.e., Eq.(2).

109 **Theorem 1.** *With a base regularizer defined as* $\psi(r) := \mathbb{E}_{r^* \sim P_0}[l_\alpha(r, r^*)]$, *the output* $r_t(\alpha)$ *of*
110 *Algorithm 1 satisfies*

$$r_t(\alpha) \in \underset{r \in [0,R]}{\arg\min} \left[ \frac{\lambda_t(t-1)}{1 - \lambda_t} \cdot \psi(r) + \sum_{i=1}^{t-1} l_\alpha(r, r^*_i) \right], \quad \forall \alpha \in [0, 1], t \ge 2. \tag{4}$$

Specifically, (i) $\psi$ is strongly convex with coefficient $\inf_{r \in [0,R]} p_0(r)$; and (ii) if $P_0$ is the uniform distribution on $[0, R]$, then $\psi$ is the quadratic function,

$$\psi(r) = \frac{1}{2R}r^2 - (1-\alpha)r + \frac{1}{2}(1-\alpha)R.$$

Next, using Theorem 1, we obtain the following *regret bound* for our CP algorithm. Here we only consider the uniform prior, and defer the case of generic priors to longer versions of this paper (the benefit of good priors can be shown using the *local norm* analysis of FTRL [Ora23, Section 7.4]).

**Theorem 2.** *Let $P_0$ be the uniform distribution on $[0, R]$. With the step size $\lambda_t = 1/\sqrt{t}$, the output of Algorithm 1 against any $r^*_{1:T}$ sequence satisfies*

$$\sum_{t=1}^{T} l_\alpha(r_t(\alpha), r^*_t) - \sum_{t=1}^{T} l_\alpha(q_{1-\alpha}(r^*_{1:T}), r^*_t) = O(R\sqrt{T}), \quad \forall \alpha \in [0,1],$$

*where $q_{1-\alpha}(r^*_{1:T})$ denotes the $(1-\alpha)$-quantile of the hindsight empirical distribution $\bar{P}(r^*_{1:T})$, and $O(\cdot)$ subsumes absolute constants.*

Let us interpret this bound. Suppose $\bar{P}(r^*_{1:T})$ is known beforehand (but the exact $r^*_{1:T}$ sequence is unknown), then for all $\alpha$, a very reasonable strategy is to predict $r_t(\alpha) = q_{1-\alpha}(r^*_{1:T})$. Theorem 2 shows that without statistical assumptions, Algorithm 1 asymptotically performs as well as this oracle in terms of the total quantile loss. Existing first-order optimization baselines are equipped with regret bounds of a similar type [BWXB23, GC24, ZBY24], but the key difference is that they require knowing $\alpha$ beforehand, whereas Algorithm 1 achieves low regret simultaneously for all $\alpha \in [0,1]$.

## 3  Discussion

**Any-$\alpha$ baseline**  Although not studied in existing works, it is actually possible to construct a nonstochastic CP algorithm from first-order optimization algorithms, without specifying a fixed $\alpha$ beforehand. The idea is simple: (i) evenly discretize the $[0,1]$ interval using a grid $\bar{A}$ of size $\sqrt{T}$; (ii) for each $\bar{\alpha} \in \bar{A}$, run a "base" CP algorithm targeting $\bar{\alpha}$; and (iii) at test time, given a queried $\alpha$, follow the base algorithm corresponding to its nearest neighbor in $\bar{A}$. It also satisfies the regret bound in Theorem 2, since the nearest-neighbor approximation only adds an additive $O(R\sqrt{T})$ factor due to the Lipschitzness of the quantile loss function $l_\alpha(r, r^*)$ with respect to $\alpha$.

However, such a baseline also suffers from the previously mentioned validity issues. Even more, the update (based on $r^*_t$) and the queries (based on $A_t$) are coupled: if $A_t$ is empty for a certain $t$ (all the users abstain), the baseline still needs $O(\sqrt{T})$ time in that round to process the observation $r^*_t$. In comparison, Algorithm 1 needs one UPDATETIME to process $r^*_t$ and $|A_t|$ QUERYTIME to answer the $\alpha$-queries, where their exact values depend on the data structure used to maintain the belief $P_t$.

**Coverage bound**  A common objective in online CP, initiated by [GC21], is to show that given a confidence level $\alpha$, the post-hoc empirical *coverage frequency* of an algorithm approaches $\alpha$, i.e.,

$$\left| \alpha - T^{-1} \sum_{t=1}^{T} \mathbf{1}[r^*_t \geq r^*_t(\alpha)] \right| = o(1).$$

Since this can be achieved by switching between $r^*_t(\alpha) = 0$ and $r^*_t(\alpha) = R$ independently of data [BGJ+22], one needs an extra objective, such as the regret (Theorem 2), to justify the validity of an online CP method. Existing first-order optimization baselines satisfy both desirable bounds.

Here we argue that the regret could be a better-posed objective than the coverage. To support this argument, notice that just like the previous pathological example, first-order optimization baselines achieve the coverage bound due to the "overshooting" provided by the loss linearization, and the latter also causes the validity issues discussed earlier. Besides, achieving the coverage bound requires adjusting the prediction based on the *coverage history*: if an algorithm keeps mis-covering, then it has to predict a very small $r_t(\alpha)$ to "almost ensure" coverage. These are different from regret minimization, where loss linearization is not necessary, and the algorithm is incentivized to best-respond to its belief (on the empirical distribution of the environment in hindsight).

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

# Appendix

## A   Bayesian interpretation

We further discuss the Bayesian interpretation of our algorithm, i.e., how the belief update Eq.(3) can be viewed from the statistical lens as the Bayesian nonparametric estimation of the underlying distribution from iid samples. We will follow [GCS$^+$21, Chapter 23]. This is not new, and we provide it only for the readers' reference.

**Distribution estimation**   Consider the following standard statistical problem: given $x_1, \ldots, x_n \in \mathcal{X}$ sampled iid from an unknown distribution $X$, what is a good estimate of $X$? The simplest nonparametric estimate is just the empirical distribution $\bar{P}(x_{1:n})$.

However, there is a parallel Bayesian perspective. It says that before observing any samples, we hold a certain *prior* $F_0$ on $X$, where $F_0$ is a distribution on all possibilities of $X$ (i.e., all distributions supported on the domain $\mathcal{X}$). Then, after observing the samples $x_{1:n}$, we can use the Bayes' theorem to compute the *posterior* $F_n$, the distribution of $X$ conditioned on the samples. Our estimate of $X$ can be just $\mathbb{E}[F_n]$, the expectation of the posterior. This is *Bayes-optimal* under the square loss.

Concretely, one would like $F_0$ to be a *conjugate prior*: it refers to a family of distributions (over $X$) such that if the prior $F_0$ belongs to this family, then the posterior $F_n$ also belongs to this family. The most notable conjugate prior for distribution estimation is the *Dirichlet process* (DP), denoted as $\mathrm{DP}(\alpha, P_0)$. Here $\alpha$ and $P_0$ are hyperparameters of a DP: $P_0$ equals the mean $\mathbb{E}[\mathrm{DP}(\alpha, P_0)]$, while $\alpha$ controls the variance of $\mathrm{DP}(\alpha, P_0)$. The larger $\alpha$ is, the smaller the variance of $\mathrm{DP}(\alpha, P_0)$ gets. Due to the conjugacy, if the prior $F_0 = \mathrm{DP}(\alpha, P_0)$, then the posterior after iid observations $x_{1:n}$ is

$$F_n = \mathrm{DP}\left(\alpha + n, \frac{\alpha}{\alpha + n}P_0 + \frac{n}{\alpha + n}\bar{P}(x_{1:n})\right).$$

Consequently, the Bayesian estimate of the distribution $X$ is

$$\mathbb{E}[F_n] = \frac{\alpha}{\alpha + n}P_0 + \frac{n}{\alpha + n}\bar{P}(x_{1:n}).$$

This is the same as the belief update Eq.(3) in our algorithm, with $\lambda_t = \alpha/(\alpha + n)$.

A more intuitive but less rigorous explanation: the Bayesian estimate $\mathbb{E}[F_n]$ could be regarded as adding "fictitious counts" to the samples $x_{1:n}$. It means that before observing $x_{1:n}$, we sample fictitious data $\tilde{x}_{1:N} \in \mathcal{X}$ from the prior $P_0$ (for some large $N$) and give each of them equal but small weights, such that their total weight equals $\alpha$. Then, after observing the true samples $x_{1:n}$, our Bayesian distribution estimate is the "weighted" empirical distribution taking both $x_{1:n}$ and $\tilde{x}_{1:N}$ into account.

**Adversarial Bayes**   Deviating from the above, a novelty of our work is rigorously showing that in an adversarial setting (without statistical assumptions), it is still beneficial to maintain the same Bayesian algorithmic belief on the environment and best-respond to that. Mathematically this is simple after establishing the downstream equivalence to regularization (Theorem 1), but the connection between this idea and CP is quite surprising to us.

To provide more context, such an idea of "adversarial Bayes" is closely related to the use of *Follow the Perturbed Leader* (FTPL) in adversarial online learning: in each round, FTPL randomly perturbs a summary of the historical observations, and best-responds to that using an optimization oracle. This can be regarded as best-responding to a belief *sampled* from a Bayesian posterior (rather than the posterior mean), and prior works on U-calibration (with possibly nonconvex losses) [KLST23, LSS24] are essentially built on this idea. Another well-known example is *Thompson sampling*, a prevalent Bayesian approach for bandits and reinforcement learning [LS20, XZ23].

Different from U-calibration and bandits, the online CP problem we consider has convex losses and *full information* feedback. This removes the need of randomization, therefore our algorithmic belief is chosen as the posterior mean. The algorithm simulates FTRL rather than FTPL on the output, which is deterministic, analytically simpler, and arguably more interpretable.

## B Omitted proofs

**Theorem 1.** *With a base regularizer defined as $\psi(r) := \mathbb{E}_{r^* \sim P_0}[l_\alpha(r, r^*)]$, the output $r_t(\alpha)$ of Algorithm 1 satisfies*

$$r_t(\alpha) \in \underset{r \in [0,R]}{\arg\min} \left[ \frac{\lambda_t(t-1)}{1-\lambda_t} \cdot \psi(r) + \sum_{i=1}^{t-1} l_\alpha(r, r_i^*) \right], \quad \forall \alpha \in [0,1], t \geq 2. \tag{4}$$

*Specifically, (i) $\psi$ is strongly convex with coefficient $\inf_{r \in [0,R]} p_0(r)$; and (ii) if $P_0$ is the uniform distribution on $[0, R]$, then $\psi$ is the quadratic function,*

$$\psi(r) = \frac{1}{2R} r^2 - (1-\alpha)r + \frac{1}{2}(1-\alpha)R.$$

*Proof of Theorem 1.* We first rewrite the base regularizer $\psi$ as

$$\psi(r) = \int_0^R l_\alpha(r, r^*) p_0(r^*) dr^*$$

$$= \alpha \int_0^r (r - r^*) p_0(r^*) dr^* + (1-\alpha) \int_r^R (r^* - r) p_0(r^*) dr^*.$$

It is twice-differentiable, with

$$\psi'(r) = \alpha \int_0^r p_0(r^*) dr^* - (1-\alpha) \int_r^R p_0(r^*) dr^* = \int_0^r p_0(r^*) dr^* - (1-\alpha),$$

and $\psi''(r) = p_0(r)$. The strong convexity statement on $\psi$ is thus clear. If $P_0$ is uniform, we have

$$\psi(r) = R^{-1} \left[ \alpha \int_0^r (r - r^*) dr^* + (1-\alpha) \int_r^R (r^* - r) dr^* \right]$$

$$= \frac{1}{2R} \left[ \alpha r^2 + (1-\alpha)(R-r)^2 \right] = \frac{1}{2R} r^2 - (1-\alpha)r + \frac{1}{2}(1-\alpha)R.$$

Next, consider the first part of the theorem. Algorithm 1 outputs

$$r_t(\alpha) = q_{1-\alpha} \left[ \lambda_t P_0 + (1-\lambda_t) \cdot \bar{P}(r_{1:t-1}^*) \right]$$

$$= \min \left\{ x; \lambda_t \int_0^x p_0(r) dr + \frac{1-\lambda_t}{t-1} \sum_{i=1}^{t-1} \mathbf{1}[r_i^* \leq x] \geq 1 - \alpha \right\}. \tag{5}$$

On the other hand, consider the optimization objective in Eq.(4), scaled by $(1 - \lambda_t)/(t - 1)$; it can be written as

$$\gamma(x) := \lambda_t \psi(x) + \frac{1-\lambda_t}{t-1} \sum_{i=1}^{t-1} l_\alpha(x, r_i^*).$$

Notice that the function $\gamma$ is continuous and right-differentiable. Taking its right-derivative, we have

$$\gamma'_+(x) = \lambda_t \left[ \int_0^x p_0(r^*) dr^* - (1-\alpha) \right] + \frac{1-\lambda_t}{t-1} \left[ (\alpha - 1) \sum_{i=1}^{t-1} \mathbf{1}[x < r_i^*] + \alpha \sum_{i=1}^{t-1} \mathbf{1}[x \geq r_i^*] \right]$$

$$= \lambda_t \int_0^x p_0(r^*) dr^* + \lambda_t(\alpha - 1) + \frac{1-\lambda_t}{t-1}(\alpha - 1)(t - 1) + \frac{1-\lambda_t}{t-1} \sum_{i=1}^{t-1} \mathbf{1}[x \geq r_i^*]$$

$$= \lambda_t \int_0^x p_0(r^*) dr^* + \frac{1-\lambda_t}{t-1} \sum_{i=1}^{t-1} \mathbf{1}[x \geq r_i^*] + \alpha - 1.$$

Comparing it to Eq.(5), we see that the output $r_t(\alpha)$ of Algorithm 1 satisfies

$$r_t(\alpha) = \min\{s; \gamma'_+(x) \geq 0\}.$$

Therefore it also satisfies the FTRL update, Eq.(4). $\qquad\square$

**Theorem 2.** *Let $P_0$ be the uniform distribution on $[0, R]$. With the step size $\lambda_t = 1/\sqrt{t}$, the output of Algorithm 1 against any $r_{1:T}^*$ sequence satisfies*

$$\sum_{t=1}^{T} l_\alpha(r_t(\alpha), r_t^*) - \sum_{t=1}^{T} l_\alpha(q_{1-\alpha}(r_{1:T}^*), r_t^*) = O(R\sqrt{T}), \quad \forall \alpha \in [0, 1],$$

*where $q_{1-\alpha}(r_{1:T}^*)$ denotes the $(1 - \alpha)$-quantile of the hindsight empirical distribution $\bar{P}(r_{1:T}^*)$, and $O(\cdot)$ subsumes absolute constants.*

*Proof of Theorem 2.* Starting from Eq.(4), we first verify that the regularizer weight $\frac{\lambda_t(t-1)}{1-\lambda_t}$ is increasing with respect to $t$ (when $t > 1$), so that the classical FTRL regret bound can be applied. To this end, define

$$h(t) := \frac{\lambda_t(t-1)}{1 - \lambda_t} = \frac{t - 1}{\sqrt{t} - 1}.$$

Taking the derivative, for all $t > 1$,

$$h'(t) = \frac{\sqrt{t} - 1 - \frac{t-1}{2\sqrt{t}}}{(\sqrt{t} - 1)^2} = \frac{t - 2\sqrt{t} + 1}{2\sqrt{t}(\sqrt{t+1} - 1)^2} = \frac{(\sqrt{t} - 1)^2}{2\sqrt{t}(\sqrt{t+1} - 1)^2} \geq 0.$$

Now, since the regularizer weight is increasing and the base regularizer $\psi$ corresponding to the uniform prior is $R^{-1}$-strongly convex, we can apply the strong-convexity-based FTRL regret bound [Ora23, Corollary 7.9] starting from $t = 2$ (and implicitly, $T \geq 2$). This yields

$$\sum_{t=2}^{T} l_\alpha(r_t(\alpha), r_t^*) - \sum_{t=2}^{T} l_\alpha(q_{1-\alpha}(r_{1:T}^*), r_t^*) \leq \frac{\lambda_T(T-1)}{1 - \lambda_T} \left[ \max_{r \in [0,R]} \psi(r) - \min_{r \in [0,R]} \psi(r) \right]$$

$$+ \frac{R}{2} \sum_{t=2}^{T} \frac{1 - \lambda_t}{\lambda_t(t-1)} g_t^2,$$

where $g_t$ is defined as

$$g_t = \begin{cases} \alpha, & r_t(\alpha) > r_t^*, \\ 1 - \alpha, & r_t(\alpha) < r_t^*, \\ 0, & r_t(\alpha) = r_t^*. \end{cases}$$

In all cases we have $g_t^2 \leq 1$. Furthermore, $\min_{r \in [0,R]} \psi(r) = \frac{1}{2}\alpha(1 - \alpha)R \geq 0$, $\max_{r \in [0,R]} \psi(r) = \frac{R}{2} \max\{\alpha, 1 - \alpha\} \leq R/2$. Therefore, plugging in $\lambda_t = 1/\sqrt{t}$ we have

$$\sum_{t=2}^{T} l_\alpha(r_t(\alpha), r_t^*) - \sum_{t=2}^{T} l_\alpha(q_{1-\alpha}(r_{1:T}^*), r_t^*) \leq \frac{R}{2} \left[ \frac{\lambda_T(T-1)}{1 - \lambda_T} + \sum_{t=2}^{T} \frac{1 - \lambda_t}{\lambda_t(t-1)} \right]$$

$$\leq \frac{R}{2} \left[ 4\sqrt{T} + \sum_{t=1}^{T-1} \frac{\sqrt{t+1}}{t} \right] = O(R\sqrt{T}).$$

Adding the instantaneous regret from the first round only results in an additional $R$ on the total regret bound. $\qquad\square$

