# OpenReview forum: "The Importance of Being Bayesian in Online Conformal Prediction"
_NeurIPS.cc/2024/Workshop/BDU — NeurIPS BDU Workshop 2024 Poster_

### Official Review · Reviewer_bgot · 2024-09-24
**Assessing the Validity and Robustness of Follow the Regularized Leader in Online Conformal Prediction**

**Rating:** 6
**Confidence:** 2

**Review:**

Summary: The paper proposes an algorithm (Follow the Regularized Leader) for online conformal prediction, which does not require the assumption of iid sample and addresses the validity issues from previous proposed baselines. Moreover, the proposed algorithm does not require knowing the quantile beforehand.

Strength: The paper contributes towards designing an improved algorithm that addresses the shortcomings of existing methods, particularly in terms of validity and robustness under adversarial settings.

Weakness: 1. The paper lacks empirical results compared to the various baselines, comparing the proposed algorithm to various baselines, which would help demonstrate its practical advantages. 2. The discussion on adversarial robustness is too brief. It remains unclear to me how the proposed algorithm can be thought of in the context of adversarial online learning and what mechanisms make it robust in such settings. A more detailed exploration would strengthen this aspect of the work.

---

### Official Review · Reviewer_zxuz · 2024-10-08
**Introducing the Bayesian prior in online conformal prediction**

**Rating:** 8
**Confidence:** 3

**Review:**

This paper proposed a method of online constructing the valid confidence sets given a black-box machine learning model. The key idea is to take the iid-based procedure and regularize its algorithmic belief by a Bayesian prior. The algorithm mixes the Bayesian prior with the empirical distribution of the data sequence to predict quantiles. The authors show that this method simulates a non-linear version of the FTRL algorithm, and discuss about the possibility of answering multiple confidence level queries online, and the convergence bound. Overall it is a clean and useful implementation of Bayesian theory relaxing the common i.i.d. assumptions.

My comments are:
1. How to choose appropriate priors is an unavoidable topic in such topics; It will be good if there are more discussions about what priors for what scenarios and applications, and their effects on the results. Are the results sensitive to the prior choice?
2. How to change the step size in the algorithm \lambda? There should be a strategy of quickly adapting it according to the shifts of data distribution.
3. If there are more applications in real-world datasets it will be better.

---

### Decision · Program_Chairs · 2024-10-09

Accept (Poster)